# UTAUT in Metaverse: An "Ifland" Case

Un-Kon Lee [1] and Hyekyung Kim [2],*

1 Department of Business Administration, The University of Suwon, Hwaseong 18323, Korea; snkon@suwon.ac.kr
2 Department of Liberal Arts, Sun Moon University, Asan 31460, Korea
* Correspondence: hkkim1@sunmoon.ac.kr; Tel.: +82-10-7216-0104

**Abstract:** Since 2021, big tech companies have been interested in metaverse platforms and services. Metaverse is the permanent, immersive mixed-reality world where people and people and people and objects can synchronously interact, collaborate, and live beyond the limitations of time and space, using avatars and the immersion-supporting devices, platforms, and infrastructures. On metaverse platforms, people can merge the real world and the virtual world. Because the metaverse has only recently begun to be studied, there are only dozens of studies on the metaverse published in qualified academic journals. There are few empirical studies on the extent to which metaverse platforms will be accommodated in the lives of information system users from an integrated perspective. Therefore, this paper aimed to empirically verify user acceptance of metaverse platforms by referring the unified theory of acceptance and use of technology (UTAUT). This study was conducted in two stages. (1) The concept and research trends of the metaverse platform were examined, and (2) the UTAUT model was introduced in "Ifland", one of the metaverse platforms, to verify its acceptance of information system users. I conducted a laboratory experiment while complying with the quarantine rules. Participants were required to watch a 15 min lecture on artificial intelligence on the metaverse platform "Ifland" for a sufficient time, then they discussed the impacts of artificial intelligence with others in the lecture. A total of 120 valid data points, excluding insincere responses, were collected, and hypotheses were verified through PLS analysis. Results indicate that performance expectancy, effort expectancy, and social influence of the metaverse platform significantly increased satisfaction, usage intention, purchase intention, and word-of-mouth intention. Facilitating conditions had no significant impact on satisfaction. The results of this study provide implications for how the metaverse platform should be designed and what factors should be emphasized to increase user acceptance of metaverse platforms.

**Keywords:** metaverse; definition of metaverse; research trend of metaverse; unified theory of acceptance and use of technology (UTAUT); Web 3.0; metaverse platform usage; metaverse item purchase intention; Ifland

## 1. Introduction

Companies and individuals have been paying keen attention to metaverse platforms and services since 2021. Metaverse is a composite word that combines meta, which means transcendence, beyond, virtual, or abstract, and universe, which means the world. Metaverse was originally used in Neal Stephenson's novel *Snow Crash* (1992) [1], which depicts a 3D virtual-reality world in which individuals in the real world use avatars. Inspired by this novel, Philip Rosedale developed Second Life in 2003, which is a free 3D virtual world where users can create, connect, and chat with others from around the world using voice and text, but this failed to gain huge popularity [2]. There may be many reasons why it was not very popular, but at that time, networks and devices had limitations in supporting 3D graphics [2,3]. In 2010, some servers were shut down and 30% of employees were fired. It has not received attention for a long time, but various recent changes, such as the spread of 5G broadband networks, the development of deep learning technology and immersive virtual-reality devices, the possibility of trading virtual goods using NFT

and cryptocurrency, and the need for non-face-to-face activities due to the COVID-19 pandemic have led to a reevaluation of the potential of the metaverse [4]. Nvidia CEO Jensen Hwang declared that the company's development direction will be to create a metaverse in April 2021 [5], and Facebook CEO Mark Zuckerberg changed Facebook's brand name to Meta in October 2021 [6]. These developments resulted in reexamining the potential of metaverse both in individual usage and from business perspectives [3].

From the perspective of individual users, people can move across the real world and the virtual world by using the metaverse platform. The movie "Ready Player One" successfully describes how people use the metaverse for vicarious satisfaction. The protagonist living in poverty and disadvantage in the real world accesses a virtual world named "Oasis" using immersive virtual-reality headsets. In the virtual world, they create avatars, make friends, and perform missions. Relationships in the virtual world affect social interactions and lives in the real world. Even impossible things in the real world are possible in the virtual world, and vicarious satisfaction is achieved by enjoying novel experiences. Among Generation Z, born after 2000, the metaverse is particularly popular because the real world's identity and the virtual world's identity are closely connected. Geppetto serves over 200 million subscribers. The number of Roblox's monthly active users (MAU) is approximately 150 million, of which 2/3 users are children aged 9–12 years in the US, and 1/3 users are under 16 years [4].

From the perspective of companies and platform developers, big tech companies have developed business models using metaverse platforms, which play various roles within the ecosystem [3,7]. They connect the values created within the metaverse to the real world [7]. The metaverse changes the way people use the Internet, the way people communicate, and the way they work [8]. Meta launched an immersive virtual-reality headset called Oculus Quest 2 and launched the virtual world platform Horizon [7]. Following the launch of Horizon Workroom Beta, which supports virtual offices, it is expanding to Horizon World, which is the extended virtual world. In Horizon Workroom Beta, people can use avatars in a virtual office to interact with others, share their data, conduct meetings, and check the progress of their work. Entertainment services are using metaverses, with 12.3 million people attending a concert (https://www.youtube.com/watch?v=wYeFAlVC8qU, accessed on 4 January 2022) held by Travis Scott within the Fortnite game run by Epic Games [7]. On the Roblox platform, people produce and sell their own content and items. That is why metaverse is sometimes called "Web 3.0" [8,9]. A metaverse ecosystem is already being created [3]. There are metaverse platforms and content providers (i.e., Roblox, Zepeto, Fortnite, Sandbox, and Decentraland), user experience (UX) designers (e.g., Tafi, Oculus), economies (e.g., Coinbase), and other infrastructures (e.g., network and cloud service providers) in the metaverse ecosystem [3]. Cryptocurrency and non-fungible tokens (NFTs) connect the virtual world and the real world as a means of value exchange [4]. Gucci and Coca-Cola are selling their NFTs in Decentraland [4]. The global eXtended Reality (XR) market is expected to grow to USD 1.5 trillion in 2030 [10].

Because the industry's movement toward utilizing the metaverse has only recently led to academic research, studies have not yet been conducted. There is little research on what the metaverse is, what areas the metaverse can be introduced into, and what motivations there are for introducing the metaverse into businesses, aside from its potential. Most of the studies on the metaverse are classified into two categories: (1) conceptualization, taxonomy, and classification of the metaverse [4,11]; and (2) business modeling and applications of the metaverse [7,8,10–13]. Studies in the first category remain in the definition or classification of metaverse as the four types of services: augmented reality, virtual reality, life logging, and mirroring [11]. The second category is the study of the areas and methods for using metaverse application in business. However, there are still only dozens of studies on the metaverse published in qualified academic journals [4,12]. These studies explained fragmentary perspectives to induce metaverse platform acceptance of users. Many studies have tried to account for the business cases but failed to identify the motives of metaverse

platform acceptance and to empirically validate the causal relationship between motives in an integrated model.

Therefore, this study aims to empirically verify the acceptability of the metaverse platform using the unified theory of acceptance and use of technology (UTAUT). The purpose of this study is to provide cornerstones for future studies by summarizing the findings of prior studies on the metaverse and to discover factors inducing metaverse usage. This study was conducted in two stages: (1) I performed a literature review on the metaverse; (2) the UTAUT model was adopted in the environment of "Ifland", one of the metaverse platforms in Korea, to verify user acceptance of the metaverse platform. This study can provide basic knowledge for subsequent studies by organizing the findings and discussions of prior studies on the metaverse. This study can also provide implications for what factors should be considered important for the metaverse to be accepted by users, and how they should be designed to serve as social platforms.

## 2. Literature Review

### 2.1. Metaverse

The metaverse is considered a new connection platform for Web 3.0 [8,9,13]. The metaverse will fundamentally change the way people communicate, interact, create value, and generate economies. Within the next decade, people will be able to experience immersive Internet platforms and dive not only into the real world but also into the virtual world [11]. The metaverse has already begun to evolve into Web 3.0, but no one has a clear explanation of what it will look like [8]. If Web 1.0 connects us online, and Web 2.0 connects us in an online community, Web 3.0 connects us in a community-owned virtual world [13]. The metaverse is at the forefront of Web 3.0's evolution [8,13]. The metaverse will generate more than USD 1 trillion in market value in the next few years [13]. Despite the potential and importance of the metaverse, very few studies are conducting research on the metaverse [9,14].

Studies on the metaverse are divided into two streams: (1) research that conceptualizes what the metaverse is and (2) research that defines how the metaverse makes changes in our lives and how the metaverse should be utilized in business [12]. First, there is no consistent explanation or consensus of prior studies on the definition of metaverse [4]. Dozens of studies have only operationalized the concept of the metaverse to suit each research context [4]. Park and Kim (2021) [4], in a remarkable review paper, summarized the definition of the metaverse mentioned in each study. I updated the review results and added some articles here, reflecting the recent development of technology and platforms. A total of 10 experts in information technology and media help to create a consistent definition. They extracted the morpheme of each definition and mapped each paper by its components. The definitions presented in a total of 64 papers were collected. The main components mentioned in each definition were avatar, world, synchronicity, interactivity, immersion and realism, social collaboration, permanence, and others. The avatar component refers to whether users can create avatars and express themselves. The world component reflects whether it is a virtual world, an augmented-reality world, or a mixed-reality world. The synchronicity component determines whether activities are conducted in real time. The interaction component determines whether people can manipulate objects in the metaverse world. The immersion and realism components determine how well the metaverse portrays the real world and how users can immerse themselves in the metaverse world. The social collaboration component determines the extent to which users can closely interact with each other on the metaverse platform. The permanence component how the metaverse platform continues. Additionally, the others component reflects recent technological advances, such as artificial intelligence, sensors, and blockchain, as well as social factors, such as life and customization. This study successfully synthesized the definitions of other studies, and in this study, the metaverse is defined as "the permanent, immersive mixed-reality world (including the virtual world as the parallel world of the real world or the real world of data being augmented) where people and people, people and objects can synchronously interact,

collaborate, and live over the limitation of time and space, using avatar, the immersion-supporting devices, platform, and infrastructure". Table 1 shows the results of reorganizing and mapping the definitions of each study.

**Table 1.** Definition of the metaverse.

| Source | Definition of the Metaverse | A | W | T | I | R | S | P | O |
|---|---|---|---|---|---|---|---|---|---|
| Stephenson (1992) [1] | A world where humans as avatars interact with each other and with software agents in a 3D space that reflects the real world. | O | V | O | O | | | | |
| Schroeder et al., (2001) [15] | A resident virtual world where the geography and physical characteristics of the real world are modeled in a networked digital space where the user is as an avatar. | O | V | | | | | | |
| Jaynes et al., (2003) [16] | An immersive environment using a universal and shared digital media network that removes the barriers of time and space by deceiving user's visual senses. | | V | O | | O | O | | |
| Ondrejka (2004) [17] | The technical challenges of making something close to the complexity and realism depicted in Snow Crash. | | | | | O | | | |
| Kemp and Livingstone (2006) [18] | Access online systems as exclusive clients and interact with content and other residents. | | V | | O | | O | | |
| Smart et al., (2007) [11] | The junction or nexus of our physical and virtual worlds. | | M | | | | | | |
| Goertzel (2007) [19] | An increasingly intelligent world where AGIs are integrated into interacting human social networks. | | | | | | O | | 1 |
| Rymaszewski et al., (2007) [20] | An environment where you can create your personality, quick visit different places, explore expansive buildings, and shop your way. | O | V | O | | | | | |
| Collins (2008) [21] | Form business to entertainment, and interactive network with continuous, immersive 3D virtual environments accessible. | | V | | O | O | O | O | |
| Wright et al., (2008) [22] | Extensive 3D network virtual world that can support many people at the same time for social interaction. | | V | O | | | O | | |
| Schlemmer et al., (2009) [23] | Extension of the parallel space of the physical world within the virtual internet space into cyberspace. | | V | O | | | | | |
| Schaf et al., (2009) [24] | A world of enhancing the feeling of being in a classroom rather than being an incorporeal observer in a 2D virtual environment. | | V | | | | O | | |
| Prisco (2009) [25] | A complete video-realistic medium based on virtual reality allows immersive interaction between participants. | | V | | O | O | O | O | |
| Messinger et al., (2009) [26] | A virtual world where thousands of people can interact simultaneously within the same simulated 3D space. | | V | O | | | O | | |
| Hazan (2010) [27] | A place where users log in all the time to interact with others in play, commerce, creativity, and exploration. | | V | | | | O | | 2 |
| Papagiannidis et al., (2010) [28] | A continuous world designed to give users control over almost every aspect of the world creating the object they want. | | V | | O | | | O | |
| Forte et al., (2010) [29] | A virtual place where an individual's cyber community can share social interactions without constraints of the physical world. | | V | O | | | O | | |
| Cunningham (2010) [30] | A compound word of meta and universe, meaning beyond, a temporal spatial aspect where the real world and virtual world are mixed. | | M | | | | | | |
| Cline (2011) [31] | A reimagined version of the OASIS in Ready Player One. | | V | | | | | | |
| Owens et al., (2011) [32] | An Immersive 3D virtual world in which people interact with each other and their environment, using real world metaphors but without physical limitations. | | V | O | O | O | O | | |
| Toneis (2011) [33] | A world that reconstructs the meaning of living world with the experience. | | V | | | | | | 2 |

**Table 1.** *Cont.*

| Source | Definition of the Metaverse | A | W | T | I | R | S | P | O |
|---|---|---|---|---|---|---|---|---|---|
| Guo et al., (2011) [34] | A computer simulation that allows avatars to interconnect and communicate in relatively lifelike environments. | O | V | | | | O | | |
| Connolly et al., (2011) [35] | Continuous online 3D world. | | V | O | | | | O | |
| Resmini et al., (2011) [36] | One of the variants of the Matrix movie with some good swordsmanship or zero gravity kung fu. | | V | | O | | | | |
| Muller (2012) [37] | A world-like electronic memory and the internet as a virtual reality where users log in every day. | | V | | | | | | |
| Xanthopoulou and Papagiannidis (2012) [38] | A 3D extension of the traditional electronic space that typically host massively multiplayer online role-playing games. | O | V | | | | O | | |
| Cameron (2012) [39] | Utopia and dystopian futures, where people live more in virtual worlds than in reality. | | V | | | | | | 2 |
| Hughes (2012) [40] | An asynchronous environment that users connect to and an avatar connected world that is a proxy for digitally represented human being. | O | V | | | | | | |
| Kim et al., (2012) [41] | A collective online space created by combining some physical reality enhanced by a 3D virtual world and a physically permanent virtual space. | | V | | | O | | O | |
| Kipper et al., (2012) [42] | Cyberspace where everyone is interconnected, similar to the internet accessed through a medium called virtual reality. | | V | | | O | | | |
| Kanematsu et al., (2013) [43] | A 3D virtual space where the avatar is activated on behalf of the users. | O | V | | | | | | |
| Kim et al., (2013) [44] | The virtual world which connects physical devices such as bio-sensors. | | V | | | | | | 3 |
| Preda et al., (2013) [45] | Collective online shared space. | | M | | | | O | | |
| Luse et al., (2013) [46] | Virtual world technology that allows you to live to your virtual life online. | | V | | | | | | 2 |
| Dionisio et al., (2013) [47] | An integrated network of 3D virtual worlds in an independent virtual world or an attractive alternative realm for human sociocultural interaction. | | V | | O | | O | O | |
| Gonzalez et al., (2013) [48] | Instantiation of a 3D virtual space where people interact with each other via avatars and clients. | O | V | | | | O | | |
| Ko and Jang (2014) [49] | An online virtual community that allows the use of simulations and objects to interact with other users through avatars. | O | V | | O | | O | | |
| Dascalu et al., (2014) [50] | New environments and visualizations where physical and digital objects coexist and interact in real time. | | V | O | O | O | | | |
| Amorim et al., (2014) [51] | An immersive environment that can simulate real world features. | | V | | O | | | | |
| Moldoveanu et al., (2014) [52] | Open 3D platform, consisting in a collection of customized 3D world. | | V | | | | | | 4 |
| Kwanya et al., (2014) [53] | Online shared space created by convergence. | | V | | | | O | | |
| Yoon et al., (2015) [54] | An immersive world of information where anything you can imagine today is connected to the internet and intensely stimulated the senses. | | V | | O | | | | 3 |
| Barry et al., (2015) [55] | A virtual 3D world where the avatar does everything for you. | O | V | | | | | | |
| Rehm et al., (2015) [56] | Virtually augmented physical reality and physically persistent virtual space. | | A | | | O | | O | |

**Table 1.** *Cont.*

| Source | Definition of the Metaverse | A | W | T | I | R | S | P | O |
|---|---|---|---|---|---|---|---|---|---|
| Chen (2016) [57] | Immersive environments that reflect the real world and are cocreated by residents using their imaginations. | | V | | | O | O | | 4 |
| Zackery et al., (2016) [58] | A world that can exist in different temporally, politically, and culturally different forms through human machine interactions enables the game's agents to solve present problems, redefine the past and invent the future. | O | V | | | | | | 3 |
| Choi and Kim (2017) [59] | A space created by the fusion of virtual reality and augmented reality as a compound world of abstract concepts meta and universe. | | M | | | | | | |
| Kanematsu et al., (2017) [60] | Created world with four different factors such as realism, ubiquity, interoperability, and extensibility. | | V | O | O | O | | | |
| Nevelsteen (2018) [61] | An interactive human–computer mediated simulation of an artificial environment as a permanent, synthetic, 3D, non game-centric space that separates games and social space. | | V | | O | | | O | O |
| Ryskeldiev et al., (2018) [62] | A constantly updated world of mixed-reality spaces mapped to different geospatial locations. | | M | | | | | | |
| PWC (2019) [10] | Immersive environments that show computer generated 360-degree video or that present digital information, objects, or media in the real world by using mobile devices and headset. | | M | | | | | | |
| Huggett (2020) [63] | A world where virtual worlds combine immersive VR with physical actors, objects, interfaces, and networks in a future form of the internet. A social virtual world the parallels and replaces the real world. | | V | | O | O | O | | |
| Suzuki et al., (2020) [64] | A world of dimensions in which the avatar acts on behalf of the user in the real world. | O | V | | | | | | |
| De Decker and Peterson (2020) [65] | The conceptual environment in which users of networked computers interact. | | | | O | | | | |
| Siyaev and Jo (2021) [66] | Stunning mixed-reality digital space inside the physical world, interacting in millions of 3D virtual experiences. | | M | | | | O | | |
| Dowling et al., (2021) [67] | A next generation virtual world built in blockchain. | | V | | | | | | 5 |
| Duan et al., (2021) [68] | The world that the users, as avatars, can interact with each other and software applications in a 3D virtual space. | O | V | | O | | O | | |
| Collins (2021) [69] | A massive virtual world where millions of people or their avatars will interact in real time. | O | V | | | | O | | |
| Caulfield (2021) [9] | A shared virtual 3D world, or worlds, that are interactive, immersive, and collaborative. | | V | | | O | O | | |
| Kim (2021) [3] | An interoperated persistent network of shared virtual environments where people can interact synchronously through their avatars with other agents and objects. | O | V | O | O | | | O | O |
| Seok (2021) [7] | A 3D virtual space where the socio-economic behaviors take place line the real world. | | V | | | O | O | | |
| Shen et al., (2021) [70] | The next generation of Internet and where interconnected, shared, and persistent 3D virtual spaces coexist. | | V | | | | O | O | |
| Kye et al., (2021) [71] | A virtual reality existing beyond reality. | | V | | | O | | | |
| Korean Government (2022) [72] | A world where people and people, people and things interact in a virtual space, where virtual world and real world are converged, and where creates social, cultural, and economic values. | | M | | O | | O | | 2 |

Abbreviations: A—avatar; W—world (V—virtual reality; A—augmented reality; M—mixed reality); T—synchronicity; I—interactivity; R—immersion and realism; S—social collaboration; P—permanence; O—others (1—artificial intelligence; 2—life; 3—sensor; 4—customization; 5—blockchain).

Second, studies on the applicability and ripple effect of metaverse in the business area were conducted. These studies are mainly industrial trend reports rather than academic papers. Some studies have summarized the findings of existing studies well [7,10–13,72]. Smart et al. (2007) [11] classified the main implementation forms of the metaverse into four categories: virtual reality, augmented reality, life logging, and mirroring. Narin (2021) [12] reviewed and collected a total of 40 qualified academic papers investigating the metaverse, including virtual reality, augmented reality, and mixed reality, conducted over the past 20 years, on Web of Science, and summarized the results of each study. The results of the study show that the study on metaverse is a study on the conceptualization of the metaverse, as well as a study explaining the areas in which the effectiveness of the metaverse has been verified. The effectiveness of the metaverse has been verified especially in arts, culture, religion, games, education, retailing, and other areas. PWC (2019) [10] predicted that the market size of the extended reality would reach USD 1.5 trillion by 2030, and they predicted that the metaverse will change human life in (1) product and service development, (2) healthcare, (3) HRD and training, (4) business process implementation, and (5) retail and customer areas. Seok (2021) [7] described major players of the metaverse ecosystem and four business models of (1) content creation, (2) media brokerage fee, (3) marketing, and (4) subscription fee in three dimensions: (1) device, (2) software, and (3) infrastructure. Grayscale research (2021) [13] reviewed the changes caused by the collapse of the walls of the real economy and the virtual economy by combining metaverse, NFT, and cryptocurrency from the perspective of monetary and financial contracts. According to Grayscale research (2021), the metaverse is expected to evolve from "Web 2.0 closed corporate metaverse which is centralized and controlled by big tech companies" to "Web 3.0 open crypto metaverse which is decentralized and controlled by global users". These Web 3.0 metaverses are part of a larger interconnected crypto cloud economy and support payment networks, decentralized finance, NFT sovereign goods, decentralized governance, decentralized cloud, and self-sovereign identity. According to the Korean government's "Metaverse Industrialization Policy" (2022) [72], the Korean government regards the metaverse as an opportunity to enhance national industrial competitiveness and is considering how to build the metaverse platform ecosystem. They intend to develop applications in 10 areas of life especially: tourism, culture and arts, education, healthcare, media, contents creation, manufacturing, office, and government.

However, despite these two categories of papers, some researchers argue that prior studies have limitations in external validity because they do not reflect the current advanced metaverse platform [4]. Park and Kim (2021) [4] argue that most of the prior studies on the metaverse before 2020 mainly focused on Second Life, and studies on the advanced metaverse platform that have evolved technically since 2020 are needed. They argue that the metaverse platforms of the two periods differ in three aspects [4]. First, metaverse services were provided using PCs in the past, but now they can be used anytime, anywhere, using mobile devices such as smartphones and immersive virtual-reality headsets. Second, the dramatic development of video processing and deep learning technology enables a more immersive environment and natural movement. Third, the boundaries between the virtual world and the real world are fading, as users can create contents and trade them with others using NFT and cryptocurrency on the metaverse platform. Fourth, COVID-19 caused people to enter quarantine, and non-face-to-face activities became necessary to continue their lives; the metaverse platform was used as a space for non-face-to-face activities [4]. The metaverse is not regarded as a mixed-reality technology, but as a social platform in which people live [4]. Therefore, they argued that new studies are needed for identifying the concept of the metaverse and verifying user acceptance of current advanced metaverse platforms [4].

In summary, a metaverse refers to the permanent, immersive, mixed-reality world where people and people and people and objects can synchronously interact, collaborate, and live over the limitation of time and space, using avatars and immersion-supporting devices, platforms, and infrastructures. The metaverse could be adopted into various

business areas, and can become the virtual economy platform, especially as it is combined with NFT and cryptocurrency. Since metaverse platforms before 2020 and current metaverse platforms are different in many aspects, some studies are needed to verify user acceptance of metaverse platforms. I contribute to the basic information by providing an in-depth review results of the previous studies on the metaverse for future studies. Subsequently, I seek to verify user acceptance of the metaverse platform using the UTAUT.

### 2.2. *The Unified Theory of Acceptance and Use of Technology (UTAUT)*

Even before the UTAUT, there were many theories explaining why people use information systems. Because theories do not sufficiently explain why people use information systems, because they only explain fragmentary parts of the reasons why people use information systems, and because there are many overlaps among the causes in the theories, several researchers argue that theories are needed to explain the acceptance of technology by information technology users from an integrated perspective [73]. Venkatesh et al., (2003) reflects these needs [74]. They integrated eight theories and models, such as TRA (theory of reasoned action), TPB (theory of planned behavior), TAM (technology acceptance model), TAM-TPB, MM (motivational model), MPCU (model of PC utilization), IDT (innovation diffusion theory), and SCT (social cognitive theory) to propose the unified theory of acceptance and use of technology (UTAUT) [74,75].

According to the UTAUT, the four independent variables influencing the information system user's behavioral intention are performance expectation, expected effort, social impact, and facilitating conditions [74]. Performance expectancy is defined as "the degree to which using a technology will provide benefits to consumers in performing certain activities" [74,75]. High performance expectancy reflects the information system user's belief that the information systems usage will be helpful in improving their work performance. Venkatesh et al., (2003) summarized the perceived usefulness, extrinsic motivation, job fit, relative advantage, and outcome expectations, among the factors identified in previous studies [74]. Effort expectancy is "the degree of ease associated with consumers' use of technology" [74,75]. It reflects how easily the information system can be used. Expected efforts were proposed to reflect the perceived ease of use, complexity, and ease of use in the prior models [74]. Social influence is "the extent to which consumers perceive that the important others (e.g., family and friends) believe they should use a particular technology" [74]. It means how an information system user believes that other people related to them expect them to use the information system. Social influence synthesizes subjective norms, social factors, and images of existing theories. Information system users may use the information system according to the recommendations or expectations of others in addition to their own motives. Facilitating conditions refer to "consumers' perceptions of the resources and support available to perform a behavior" [75]. It is the acknowledgement from information system users of the existence of several supporting structures which allow users to use the information system smoothly [75]. Facilitating conditions are variables that combine perceived behavioral control, facilitating conditions, and compatibility of existing theories [74]. When information system users believe that they have sufficient conditions to use the information system, they intend to use the information system. Behavioral intention is the willingness to use the information system, which leads to actual behaviors [74]. The path through which the four independent variables influence behavioral intention is moderated by four factors: gender, age, experience, and voluntariness of use [74].

Originally, they argued that four independent variables directly influence behavioral intention in the UTAUT. However, according to the TPB (theory of planned behavior) referred to in the UTAUT, it is argued that the cognitive and affective perception of information system users first forms an overall attitude toward the information system and then affects the behavioral intentions [76]. Therefore, it is reasonable to judge that these four independent variables influence the formation of attitudes of information system users and then affect behavioral intentions. One of the most widely used variables among attitudes is the overall satisfaction that information system users feel about using the information

system [77]. Service satisfaction is defined as the overall positive evaluation of toward the service usage by users [78]. When users could feel positive about their experience using the information system, they could be satisfied with the information system usage and continue to use the information system to keep their positive experience [77]. Consumers' purchase intentions are influenced by satisfaction [78,79]. When users are satisfied with the information system, they want to return the corresponding value to the information system service provider [80]. In this case, they could spread positive word of mouth, sharing their positive experiences with others [80].

The UTAUT has been adopted as the overarching theory in more than 800 papers since 2003 [81]. The UTAUT has also been demonstrated in a variety of contexts, including online discussion forums, online shopping, mobile banking, RFID, E-government, and even E-leaning [81]. Therefore, I establish hypotheses using the UTAUT to verify users acceptance of the metaverse in this study.

I controlled the exogenous variables, such as gender, age, experience, and voluntariness of use, which are the moderating variables in the UTAUT. This is because I wanted to clearly observe the influence of the four independent variables of the UTAUT. Media richness [82,83], which is the main motivation for media users to use a media, and information overload [84], which expresses the difficulty of information and acceptability of information consumers, were also controlled in this study.

## 3. Hypotheses Development and Research Model

Since the early studies on the metaverse mainly targeted Second Life, they did not reflect the characteristics of the current, advanced metaverse platform. (1) Broadband convergence networks, (2) development of immersive, virtual-reality-supporting technologies and devices, (3) transactions using NFT and cryptocurrency, and (4) increasing demand for non-face-to-face activities due to the COVID-19 pandemic accelerated the evolution of the metaverse platform. Research is needed on how information system users will accept this advanced metaverse platform. Since the metaverse platform is also a form of information system, it is possible to introduce the UTAUT to explain, from an integrated perspective, why users accept the metaverse platform.

First, performance expectancy indicates the degree to which information system users believe that the use of information systems will help improve their work performance. Using broadband convergence networks and the immersive, virtual-reality-supporting technology, metaverse platform users can enjoy a vivid and immersive media experience that incumbent media has not provided. Metaverse platform users can communicate accurately and quickly to a degree similar to face-to-face communication situations. Accuracy and speed of communication increase mutual understanding between people and enable them to handle tasks effectively. When metaverse platform users can understand information accurately, they can remember the information better and make more accurate decisions for performing tasks. That will improve the work productivity of metaverse users. Metaverse platform users report positive experiences about the metaverse platform, say it is useful, and their work productivity improves: they are generally satisfied with the metaverse platform usage experiences. Therefore, Hypothesis 1 was set as follows.

**Hypothesis 1.** *Performance expectancy could positively affect user satisfaction.*

Second, metaverse platform users can use platforms more easily and handle their tasks more easily. The metaverse is a world created by imitating reality as it is, so the behavioral guidelines and rules of the real world can be applied similarly. Compared with a "completely newly created world" with no prior experience or standards, the metaverse platform helps users enjoy the metaverse world more easily. Metaverse platforms support easier communication and interaction between people and people, and between people and computers. Information on the metaverse platform is intuitive and easily understood.

Metaverse platform users are more likely to be satisfied with easily using the metaverse platform. Therefore, Hypothesis 2 was set as follows.

**Hypothesis 2.** *Effort expectancy could positively affect user satisfaction.*

Third, social influence means the degree to which people related to the user believe that the user will use the metaverse platform. Older generations were sometimes negative about use of information technology. It is natural for Generation Z to use information technology from their growth period. They create UCCs by themselves, share them with their friends, use social networks, and communicate with other friends. The boundaries between the real world and the virtual world are breaking down as transactions between buying and selling virtual products are allowed using NFT and cryptocurrency. The distinction between the self in the real world and the self in the virtual world is blurred in Generation Z. For them, it is natural to use the metaverse platform, and it is a channel recognized by other colleagues. Therefore, as they feel the metaverse is essential to form relationships with friends, the metaverse platform helps to increase their satisfaction. Therefore, Hypothesis 3 was set as follows.

**Hypothesis 3.** *Social influence could positively affect user satisfaction.*

Fourth, the development of information technology and the COVID-19 pandemic increased the demand for non-face-to-face activities and accelerated the introduction and development of metaverse platforms. Due to the prolonged COVID-19 pandemic, people went into quarantine, and they began to actively introduce immersive media platforms to replace real-life activities. These are the reasons why Nvidia and Meta have recently recognized and actively introduced the metaverse as Web 3.0. As more subscribers work on the network, the value created by the network increases exponentially. Therefore, as the metaverse platform is developed, people become more smoothly active on the metaverse platform, gain more value on the metaverse platform, and become more satisfied with using it. Therefore, Hypothesis 4 was set as follows.

**Hypothesis 4.** *Facilitating conditions could positively affect user satisfaction.*

According to prior studies on information system user behavior, satisfaction causes various positive behavioral intentions. Since metaverses are a form of information system, users who are satisfied with using the metaverse platform will create positive behavioral intentions. Users who are satisfied with using the metaverse platform want to continue this positive experience and continue to use the service [77]. Consumers who are satisfied with the metaverse platform will be willing to purchase virtual assets, such as items and contents, as well as real-world goods such as devices, software, and infrastructure to use the metaverse platform [78,79]. When metaverse platform users are satisfied, users want to share their positive experiences with other friends, and positive word-of-mouth intentions increase. Therefore, Hypotheses 5–7 are set as follows. Figure 1 shows the research model reflecting my hypotheses.

**Hypothesis 5.** *Satisfaction could positively affect the usage intention.*

**Hypothesis 6.** *Satisfaction could positively affect the purchase intention.*

**Hypothesis 7.** *Satisfaction could positively affect the word-of-mouth intention.*

Figure 1 shows the research model reflecting my hypothesis.

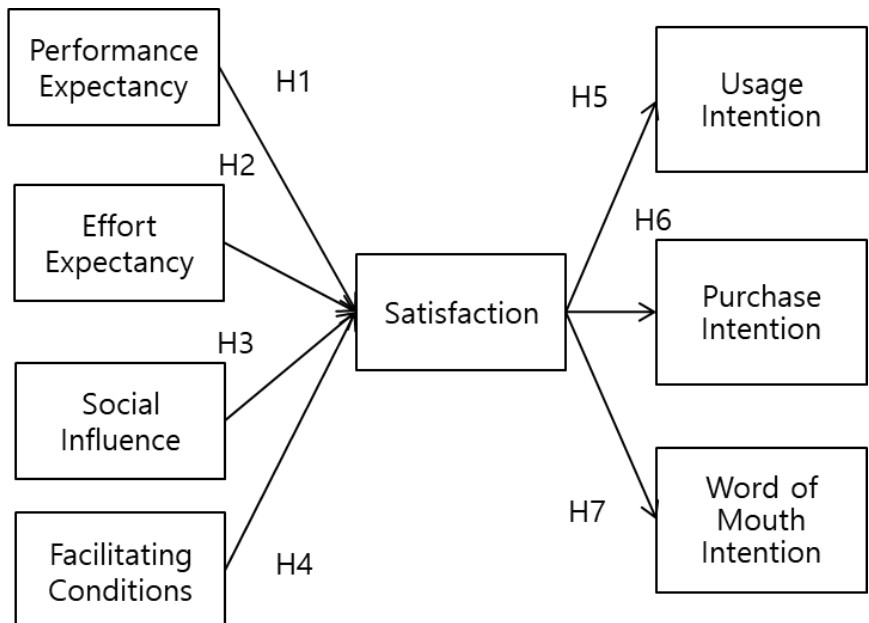

cf. Control variables: Media richness, Information overload

**Figure 1.** Research model.

## 4. Methodology

### 4.1. Operationalization of Constructs

Hundreds of prior studies have already used the measurement items of Venkatesh et al. (2003) [81]. For comparison with the results of these studies, I also used the measurement items of Venkatesh et al. (2003) [74]. However, some measurement items were dropped during the validation process. The validation of measurement items and survey questionnaires was carried out according to the three-step validation guidelines proposed by Churchill (1979) [85]. First, I used all survey questionnaires which were already validated in Venkatesh et al. (2003). Second, since the survey questionnaires I used were translated from English into Korean, I checked whether all the questionnaires express the meaning of the original question well. With the help of three experts in English and information systems, it was confirmed whether the translation was accurately performed. They carefully reviewed each questionnaire in consideration of the nuances of the sentences and words and said that all survey questionnaires were properly written. Third, I checked whether all survey questionnaires could be easily understood and answered by respondents, clearly conveyed meaning, and were appropriate for our research situation. To confirm these, I recruited 10 students, who participated in the pilot test. They argued that some questions did not fit our research situation and conveyed meanings too similar to the other questions. These were the fourth question of performance expectancy, "If I use the system, I will increase my chances of getting a raise", the first question of effort expectancy, "My interaction with the system would be clear and understandable", and the fourth question of facilitating conditions, "A specific person is available for assistance with system difficulties". Im et al. (2011) [86] also conducted their research excluding this question. Therefore, I excluded these questions in my study. Pilot test participants said the rest of the questions were all clear in meaning and were easy to answer. Table 2 shows the final, confirmed measurement items and survey questionnaires.

**Table 2.** Measurement items and survey questionnaires.

| Constructs | Measurement Items | Sources |
|---|---|---|
| Performance expectancy (PE) | (PE1) I would find the MP useful in my job.<br>(PE2) Using the MP enables me to accomplish tasks more quickly.<br>(PE3) Using the MP increases my productivity. | Venkatesh et al., (2003) [74] and Im et al., (2011) [86] |
| Effort expectancy (EE) | (EE1) It would be easy for me to become skillful at using the MP.<br>(EE2) I would find the MP easy to use.<br>(EE3) Learning to operate the MP is easy for me. | |
| Social influence (SI) | (SI1) People who influence my behavior think that I should use the MP.<br>(SI2) People who are important to me think that I should use the MP.<br>(SI3) The senior management of this business (my school) has been helpful in the use of the MP.<br>(SI4) In general, people around me have supported the use of MP. | |
| Facilitating conditions (FC) | (FC1) I have the resources necessary to use the MP.<br>(FC2) I have the knowledge necessary to use the MP.<br>(FC3) The MP is not compatible with other systems I use. | |
| Satisfaction (SAT) | The experience using this MP . . .<br>(SAT1) I am contented with.<br>(SAT2) I am satisfied with.<br>(SAT3) It meets what I expect for this type of service. | Kim and Son, (2009) [80] |
| Usage intention (UI) | (UI1) I intend to use the MP in the next 12 months.<br>(UI2) I predict I would use the MP in the next 12 months.<br>(UI3) I plan to use the MP in the next 12 months. | Venkatesh et al., (2003) [74] |
| Purchase intention (PI) | If you subscribe MP or purchase items (skin, accessories, etc.) to actively use this MP . . .<br>(VI1) The probability of subscription or purchasing in this MP would be probable.<br>(VI2) The likelihood of subscription or purchasing is highly likely.<br>(VI3) My willingness to subscribe MP or purchase items is highly willing.<br>(VI4) The probability that I would consider subscribing MP or purchasing items is highly probable | Song and Zahedi (2005) [87] |
| Word-of-mouth intention (WM) | (WM1) I will say positive things about this MP to other people<br>(WM2) I will recommend this MP to anyone who seeks my advice.<br>(WM3) I will refer my acquaintances to this MP. | Kim and Son (2009) [80] |
| Media richness (MR) | (MR1) This MP provides the MPC which could be easily understood.<br>(MR2) This MP help me to understand the MPC.<br>(MR3) This MP could not get in the way of understanding the MPC.<br>(MR4) I could easily explain the MPC (dropped).<br>(MR5) This MP helped me understand the MPC quickly.<br>(MR6) This MP could provide the various cues which help me easily to understand the MPC.<br>(MR7) This MP could provide the various cues which help me to better understand the MPC.<br>(MR8) This MP could provide the various cues which help me to quickly understand the MPC. | Kahai and Cooper (2003) [83] |
| Information overload (IO) | (IO1) I need more time to understand this MPC.<br>(IO2) This MPC contains too complex information for me to understand.<br>(IO3) This MPC contains too much information for me to understand. | Paul and Nazareth (2010) [84] |
| Fantasization (FN) | (FN1) I daydream a lot.<br>(FN2) When I go to the movies, I find it easy to lose myself in the firm.<br>(FN3) I often think of what might have been. | Kim and Son (2009) [80] |

cf. MP means metaverse platform and MPC means contents in this metaverse platform. cf2. Media richness and information overload are two control variables and fantasization is the marker variables for checking common method bias.

*4.2. Data Collection*

The population of this study comprised potential users of the metaverse platform, who have never used the metaverse before. I recruited laboratory experiment participants from students taking my class. All students agreed to participate in the experiment and received coffee coupons worth USD 10 in return. Out of the 130 participants, 5 were absent from the experiment due to personal reasons on the day of the laboratory experiment. Five people were excluded as outliers because they gave unfaithful answers to the questionnaire (all answers are 1 or 7). I gathered a total of 120 valid data points.

This study conducted laboratory experiments while following COVID-19 quarantine rules to verify the hypotheses. The procedure of the laboratory experiment was as follows. First, a brief description and explanation about the operation method of the metaverse platform "Ifland" were provided. Second, a 15 min lecture on artificial intelligence was conducted in a discussion room opened in Ifland. Third, students freely discussed with other students how the new technology of artificial intelligence will change our daily lives, and then presented the results of the discussion to other students in each group. Fourth, they were required to answer the questionnaire after using Ifland for a sufficient time. Since no more than six people could gather due to the COVID-19 quarantine rules, the students participated in the experiment in groups of five. The demographic data of respondents are shown in Table 3.

**Table 3.** Demographic data of the respondents.

| G | Num | % | W | Num | % | H | Num | % |
|---|---|---|---|---|---|---|---|---|
| MN | 62 | 51.7% | 0–1 | 33 | 27.5% | 0–1 | 21 | 17.5% |
| FE | 58 | 48.3% | 1–2 | 4 | 3.3% | 1–2 | 35 | 29.2% |
| M | Num | % | 2–3 | 6 | 5.0% | 2–3 | 31 | 25.8% |
| B | 107 | 89.2% | 4–5 | 17 | 14.2% | 4–5 | 24 | 20.0% |
| O | 13 | 10.8% | 6–7 | 22 | 18.3% | 6–7 | 5 | 4.2% |
| | | | 8–10 | 22 | 18.3% | 8–10 | 4 | 3.3% |
| | | | Over 10 | 16 | 13.3% | Over 10 | 0 | 0.0% |

cf. G—gender; M—major; W—web experience (years); H—web usage per a day (hours); Num—frequency; MN—male; FE—female; B—business; O—others.

## 5. Analysis and Results

I analyzed the data and verified the hypotheses of this study. The PLS algorithm was used for data analysis. The PLS algorithm is a technique that can proceed with analysis with fewer observations than other structural equation models of LISREL or AMOS [88]. It is suitable for analyzing the relatively few observations in this study (120). To verify the hypothesis using the PLS algorithm, two steps were performed: (1) verifying the validity of the measurement model, and (2) calculating the path coefficient between each variable [88].

*5.1. Measurement Model*

I could confirm the facial validity, convergent validity, and reliability of the measurement model by checking some statistic values and criteria. To check the reliability of each variable, I calculated the average variance extracted (AVE), composite reliability, and Cronbach's alpha [89]. All the AVE values were greater than the 0.5 cutoff, all composite reliability values were greater than the 0.7 cutoff, and all Cronbach's alpha values were greater than the 0.7 cutoff [89,90]. Two validation options could be adopted to confirm the discriminant validity of the measurement model. One option is comparing the square root value of AVE with the latent variable correlation value with the other variables. Table 4 shows that the square root values of the AVE of all variables exceeded the correlation value with the other variables. The other option is to check whether the factor-loading value in each variable is sufficiently greater than the factor-loading values between variables. Table 5 shows that the factor-loading values within each variable exceeded those

between variables. Reflecting the results of a comparison between the statistics and criteria, and two results of checking the discriminant validity in this study, I confirmed that my measurement model reached a sufficient level of validity and reliability. Then, I proceeded with a structural equation model analysis using the PLS algorithm.

**Table 4.** Latent variables correlations.

|  | AVE | CR | R2 | $\alpha$ | PE | EE | SI | FC | SAT | UI | PI | WoM | MR | IO |
|---|---|---|---|---|---|---|---|---|---|---|---|---|---|---|
| PE | 0.791 | 0.919 | 0.137 | 0.870 | 0.889 | | | | | | | | | |
| EE | 0.784 | 0.916 | 0.502 | 0.862 | 0.439 | 0.885 | | | | | | | | |
| SI | 0.750 | 0.923 | 0.277 | 0.889 | 0.438 | 0.463 | 0.866 | | | | | | | |
| FC | 0.686 | 0.867 | 0.214 | 0.769 | 0.245 | 0.620 | 0.537 | 0.828 | | | | | | |
| SAT | 0.919 | 0.972 | 0.563 | 0.956 | 0.554 | 0.495 | 0.682 | 0.452 | 0.959 | | | | | |
| UI | 0.944 | 0.981 | 0.529 | 0.970 | 0.449 | 0.401 | 0.685 | 0.367 | 0.726 | 0.972 | | | | |
| PI | 0.840 | 0.955 | 0.258 | 0.939 | 0.381 | 0.114 | 0.447 | 0.256 | 0.507 | 0.596 | 0.917 | | | |
| WoM | 0.894 | 0.962 | 0.567 | 0.940 | 0.566 | 0.490 | 0.731 | 0.424 | 0.747 | 0.883 | 0.557 | 0.946 | | |
| MR | 0.694 | 0.940 | | 0.925 | 0.336 | 0.331 | 0.514 | 0.334 | 0.557 | 0.537 | 0.387 | 0.571 | 0.833 | |
| IO | 0.779 | 0.913 | | 0.860 | −0.250 | −0.693 | −0.280 | −0.372 | −0.414 | −0.220 | −0.104 | −0.294 | −0.347 | 0.882 |

cf. AVE—average variance extracted; CR—composite reliability; $\alpha$—Cronbach's alpha. cf2. Diagonal cells are square root of AVE of each construct. Off-diagonal cells are the correlations. For the abbreviation of each variable, refer to Table 2.

**Table 5.** Factor-loading values.

| Index | PE | EE | SI | FC | SAT | UI | PI | WoM | MR | IO |
|---|---|---|---|---|---|---|---|---|---|---|
| PE1 | 0.881 | 0.494 | 0.395 | 0.297 | 0.583 | 0.480 | 0.347 | 0.605 | 0.310 | −0.324 |
| PE2 | 0.894 | 0.266 | 0.334 | 0.175 | 0.380 | 0.305 | 0.389 | 0.385 | 0.290 | −0.119 |
| PE3 | 0.892 | 0.369 | 0.429 | 0.158 | 0.475 | 0.379 | 0.284 | 0.478 | 0.291 | −0.184 |
| EE1 | 0.455 | 0.857 | 0.367 | 0.466 | 0.448 | 0.379 | 0.117 | 0.429 | 0.250 | −0.667 |
| EE2 | 0.388 | 0.894 | 0.365 | 0.580 | 0.442 | 0.291 | 0.066 | 0.427 | 0.311 | −0.592 |
| EE3 | 0.314 | 0.905 | 0.503 | 0.608 | 0.420 | 0.392 | 0.119 | 0.444 | 0.322 | −0.572 |
| SI1 | 0.393 | 0.509 | 0.831 | 0.479 | 0.594 | 0.523 | 0.218 | 0.641 | 0.426 | −0.246 |
| SI2 | 0.440 | 0.419 | 0.896 | 0.437 | 0.664 | 0.718 | 0.489 | 0.749 | 0.493 | −0.289 |
| SI3 | 0.273 | 0.262 | 0.840 | 0.414 | 0.505 | 0.533 | 0.439 | 0.516 | 0.408 | −0.128 |
| SI4 | 0.393 | 0.396 | 0.895 | 0.532 | 0.584 | 0.578 | 0.398 | 0.600 | 0.445 | −0.287 |
| FC1 | 0.218 | 0.333 | 0.393 | 0.779 | 0.420 | 0.313 | 0.264 | 0.408 | 0.286 | −0.152 |
| FC2 | 0.211 | 0.544 | 0.547 | 0.880 | 0.363 | 0.302 | 0.230 | 0.342 | 0.226 | −0.321 |
| FC3 | 0.181 | 0.660 | 0.393 | 0.822 | 0.337 | 0.296 | 0.142 | 0.303 | 0.314 | −0.448 |
| SAT1 | 0.563 | 0.476 | 0.652 | 0.376 | 0.969 | 0.738 | 0.511 | 0.744 | 0.565 | −0.396 |
| SAT2 | 0.544 | 0.514 | 0.670 | 0.465 | 0.972 | 0.683 | 0.500 | 0.712 | 0.532 | −0.451 |
| SAT3 | 0.483 | 0.432 | 0.641 | 0.462 | 0.935 | 0.665 | 0.446 | 0.691 | 0.503 | −0.342 |
| UI1 | 0.442 | 0.413 | 0.733 | 0.369 | 0.726 | 0.973 | 0.595 | 0.874 | 0.557 | −0.252 |
| UI2 | 0.412 | 0.326 | 0.618 | 0.342 | 0.676 | 0.967 | 0.598 | 0.838 | 0.490 | −0.163 |
| UI3 | 0.453 | 0.425 | 0.642 | 0.359 | 0.712 | 0.975 | 0.546 | 0.860 | 0.515 | −0.224 |
| PI1 | 0.372 | 0.179 | 0.489 | 0.331 | 0.559 | 0.672 | 0.942 | 0.629 | 0.418 | −0.072 |
| PI2 | 0.294 | 0.125 | 0.510 | 0.252 | 0.535 | 0.622 | 0.951 | 0.589 | 0.376 | −0.049 |
| PI3 | 0.389 | 0.057 | 0.252 | 0.155 | 0.336 | 0.387 | 0.888 | 0.346 | 0.264 | −0.181 |
| PI4 | 0.370 | 0.000 | 0.293 | 0.136 | 0.345 | 0.398 | 0.883 | 0.371 | 0.318 | −0.127 |

**Table 5.** *Cont*.

| | | | | | | | | | | |
|-----|--------|--------|--------|--------|--------|--------|--------|--------|--------|--------|
| WM1 | 0.506 | 0.502 | 0.673 | 0.465 | 0.707 | 0.781 | 0.469 | 0.907 | 0.508 | −0.317 |
| WM2 | 0.560 | 0.400 | 0.691 | 0.353 | 0.698 | 0.854 | 0.570 | 0.968 | 0.541 | −0.200 |
| WM3 | 0.540 | 0.486 | 0.708 | 0.382 | 0.712 | 0.868 | 0.541 | 0.960 | 0.571 | −0.314 |
| MR1 | 0.308 | 0.303 | 0.397 | 0.301 | 0.464 | 0.422 | 0.370 | 0.469 | 0.851 | −0.361 |
| MR2 | 0.164 | 0.308 | 0.432 | 0.292 | 0.484 | 0.457 | 0.255 | 0.491 | 0.851 | −0.380 |
| MR3 | 0.255 | 0.204 | 0.394 | 0.342 | 0.428 | 0.354 | 0.329 | 0.384 | 0.759 | −0.285 |
| MR5 | 0.371 | 0.373 | 0.487 | 0.293 | 0.527 | 0.512 | 0.327 | 0.573 | 0.914 | −0.283 |
| MR6 | 0.235 | 0.227 | 0.451 | 0.274 | 0.459 | 0.425 | 0.342 | 0.455 | 0.850 | −0.202 |
| MR7 | 0.314 | 0.230 | 0.252 | 0.119 | 0.312 | 0.300 | 0.173 | 0.301 | 0.686 | −0.345 |
| MR8 | 0.307 | 0.265 | 0.530 | 0.291 | 0.531 | 0.603 | 0.418 | 0.595 | 0.896 | −0.207 |
| IO1 | −0.284 | −0.526 | −0.250 | −0.248 | −0.511 | −0.303 | −0.198 | −0.344 | −0.443 | 0.864 |
| IO2 | −0.278 | −0.746 | −0.353 | −0.429 | −0.385 | −0.244 | −0.093 | −0.328 | −0.297 | 0.942 |
| IO3 | −0.049 | −0.504 | −0.069 | −0.266 | −0.166 | 0.020 | 0.037 | −0.043 | −0.163 | 0.839 |

I gathered data using the questionnaire measuring the items both of the independent variables and the dependent variables, and common method bias (CMB) was calculated. CMB is an error in which the correlation between the variables appears to be more exaggerated than it really is when both independent and dependent variables are measured in the same way from the same respondent. To check the level of CMB in this study, I adopted the marker variable technique. The marker variable technique is a method for calculating the average correlation coefficient between the variables in hypothesis and the theoretically unrelated marker variable (i.e., fantasization) [80]. To confirm the absence of CMB, the average correlation coefficient should be less than 0.10. Because the average correlation coefficient between variables and the marker variable was close to 0 ($r = 0.095$, not significant), I could confirm that CMB was not serious.

*5.2. PLS Analysis Results*

I performed structured equation model analysis using the PLS algorithm and calculated the path coefficients and significances between variables to verify my hypotheses. Path coefficients between performance expectancy and satisfaction (H1, $β = 0.279$ ***), between effort expectancy and satisfaction (H2, $β = 0.123$ **), between social influence and satisfaction (H3, $β = 0.481$ ***), between satisfaction and usage intention (H5, $β = 0.726$ ***), between satisfaction and purchase intention (H6, $β = 0.507$ ***), and between satisfaction and positive word-of-mouth intention (H7, $β = 0.747$ ***) were significant. Therefore, all the above hypotheses were significantly approved. However, the path coefficient between the facilitating condition and satisfaction (H4, $β = 0.042$) was not significant, and H4 was rejected. Figure 2 shows the PLS analysis results, which show the path coefficients and *p*-value of each path.

The interpretation of this analysis result is as follows. The more that people feel that they can use a metaverse platform to perform tasks faster and increase their work productivity, the more satisfied they are with metaverse services. People are satisfied with the metaverse service only when the metaverse is easily usable. The metaverse is a new information technology that has not existed before, so people are satisfied only when they can easily learn how to use it. A user becomes more satisfied with using a metaverse platform when people around them, those who influence their behavior, or those who they value, expect them to use the metaverse platform. However, the current metaverse service seems not to provide unique resources or services that are differentiated from other information system services. Otherwise, as the original model of Venkatesh et al. (2003) [74] argues, the facilitating condition may directly affect behavior, but not behavioral intention. I will explain the feelings and evaluations of the participants for each independent variable again in the results of the in-depth interview. The more that people are satisfied with the metaverse platform and service, the more positive behavioral intentions people form. People use a metaverse platform to maintain such experiences. If a user should need to pay subscription fees or item purchase fees for their active participation in the metaverse

platform, they are willing to pay these. A positive experience in a metaverse platform not only increases an individual's behavioral intention, but also allows them to increase the participation of other potential users by spreading positive word of mouth about the metaverse platform to others. Therefore, this study successfully verified those factors which make people accept the metaverse platform, especially the four independent variables which influence satisfaction and various behavioral intentions.

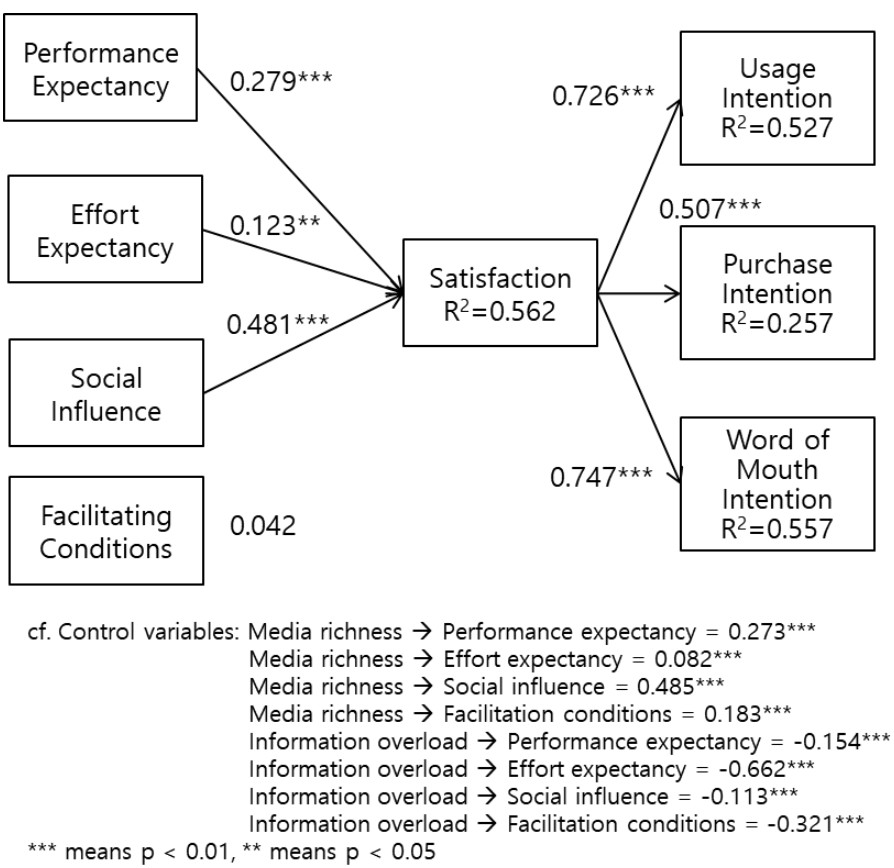

cf. Control variables: Media richness → Performance expectancy = 0.273***
                              Media richness → Effort expectancy = 0.082***
                              Media richness → Social influence = 0.485***
                              Media richness → Facilitation conditions = 0.183***
                              Information overload → Performance expectancy = -0.154***
                              Information overload → Effort expectancy = -0.662***
                              Information overload → Social influence = -0.113***
                              Information overload → Facilitation conditions = -0.321***
         *** means p < 0.01, ** means p < 0.05

**Figure 2.** PLS analysis results.

An interesting part of the analysis is to compare the size of the path coefficient of each path. Prior studies argue that, among the four independent variables of the UTAUT, performance expectancy has the greatest influence on behavioral intentions. However, the analysis results indicate that social influence had the greatest influence on satisfaction in this study. The path coefficient of effort expectancy on satisfaction was relatively small. Facilitating conditions did not significantly affect satisfaction. Satisfaction significantly increased usage intention, purchase intention, and word-of-mouth intention, but satisfaction had a relatively small impact on the purchase intention among the three behavioral intentions.

To interpret the above results, I conducted additional interviews with participants in the laboratory experiment. The four independent variables of this study, performance expectancy, effort expectancy, social influence, and facilitating conditions, proposed by the UTAUT, are abstract and broad concepts, rather than being specific to the situation. Interview results explain how participants felt about these four factors in the metaverse context. First, interview results explain that there are two reasons why social influence has the greatest influence on satisfaction. (1) Unlike the incumbent information system, introduced to efficiently perform tasks, the metaverse is a world in itself. So, the reason for its introduction is not only to increase work productivity, but also to facilitate interaction and living with people in that world. (2) Most of the subjects in this study were people aged 20–30 years, and most of the incumbent metaverse platform users are under the age of 18 years. Their behaviors are easily influenced by their friends and colleagues. Second,

the interview results explain why effort expectancy had a small impact on satisfaction. The metaverse provides information very intuitively. Metaverse users could use the metaverse very easily because rules and guidelines are similar to those in the real world. There is no need to learn specific knowledge about the metaverse. My participants already had basic knowledge for using the web interface through having had years of web experience, and this knowledge allowed them to use the metaverse sufficiently. Easy use of the metaverse is a natural requirement and a necessary condition for metaverse services. Third, the UTAUT argues that the facilitating conditions directly affect the behavior itself, not the behavioral intention. Other prior studies mainly measured the information systems usage time as a proxy of actual behavior, but this study did not measure the metaverse usage time for capturing actual behaviors, because I allowed participants to use the metaverse platform for a sufficient time. Interview results differently explain why the facilitating conditions could not significantly affect user satisfaction with the findings of prior studies. The metaverse platform of this study can be used in all smartphones, PCs, and immersive virtual-reality headsets, but is especially easy to use on smartphones. Participants were of Generation Z, and were familiar with smartphones. It is also natural for them to be compatible with other information systems using smartphones. In addition, the natural features of the metaverse could not increase their satisfaction. Interviewees answered that it was not the facilitating conditions, but the constraint conditions of the incumbent metaverse platform. They responded that the functions of Ifland, the metaverse platform used in this study, were often already provided by other games or web services, and they could not find unique functions of the metaverse platform which were different to other information systems. They responded that, since mere 3D graphics are already provided in many games, it would be better to have features that can create unique experiences of the metaverse, such as various immersive virtual-reality and interactivity support functions. Fourth, the interview results also explain why the path coefficient of purchase intention was relatively small compared with the usage intention and the word-of-mouth intention. Participants had already experienced free service during the open beta period, charging for only some premium services in numerous online games. Respondents said that they pay only when they think that the metaverse platform provides significant value for users, and that they must charge after actually using the metaverse services.

In summary, the results of this study indicate that three independent variables, performance expectancy, effort expectancy, and social influence, could increase metaverse user satisfaction and various behavioral intentions, such as usage intention, purchase intention, and word-of-mouth intention. Social influence could be essential to attract potential metaverse users.

## 6. Discussion

The more that individuals pursue vicarious satisfaction in the virtual world using metaverses, and the more big tech companies start developing metaverse platforms and business models, the more interest people have in metaverses. However, there are few studies on what the metaverse is and what factors lead to user acceptance of the metaverse. This study paid attention to the absence of these studies and reviewed prior studies to define the metaverse and summarize the flows of research. This study successfully verified four independent variables that affect the acceptance of metaverse using the UTAUT, a unified perspective on technology acceptance. Performance expectancy, effort expectancy, and social influences for the metaverse increase the satisfaction of using the metaverse platform, and significantly increase the usage intention, purchase intention, and word-of-mouth intention. This result provides several topics for discussion.

The first–third topics cover the reasons why researchers should study the metaverse. First, the metaverse is more than just virtual-reality technology. Recall that metaverses are influenced by other values, such as social influence, along with performance expectancy. Unlike other information systems, developed to effectively perform a specific task, the metaverse in itself means a world. From the perspective of media technology, I can explain

the characteristics of the metaverse as follows. In the real world, it is impossible for countless people to communicate and cooperate with each other at the same time. It is because there are time and space constraints in the real world. Media has evolved beyond these constraints to support rich and smooth communication comparable to face-to-face communication in the real world. Metaverse now means more than just a means of communication. The performance of the metaverse is an achievement in the lives of users living in it. As non-face-to-face activities increase, in-metaverse activities will also increase. The subjects of metaverse research should not be limited to verifying the effectiveness of information systems but should evolve into research that recognizes the metaverse as another world and observes the dynamics between people and people and between people and objects.

Second, the metaverse has the meaning of a fusion of the real world and the virtual world. Recall that the second flow of the metaverse definitions summarized in this study is the fusion of the real world and the virtual world. Human relationships formed through activities in the metaverse also affect human relations in the real world. Outputs of activities in the metaverse are converted into real-world value through NFTs and cryptocurrency. The real world and the virtual world now evolve into a relationship in which one world influences the other: it is not a one-way influence relationship. When the virtual world and the real world are connected in both directions, research is needed on how it will affect human life and what problems can appear in both worlds.

Third, it could be a paradigm shift through the virtualization of social relations. Recall the conclusion that social influences had the greatest influence on the acceptance of the metaverse. Even now, people spend a lot of their daily lives online. The more smoothly people interact with others in the metaverse, the more people use the metaverse. Generation Z especially sometimes has an ambiguous boundary between online identity and offline identity. They make relationships offline with people they meet online, and vice versa, offline friends sometimes have relationships online. When social relations are virtualized, problems such as crimes that abuse anonymity are also found, along with several advantages. To promote the use of metaverse platforms, research on how to use social influences is needed. Research is needed on both the positive and the negative aspects of social activity in the metaverse.

The fourth–ninth topics are discussions on how the metaverse should be designed. Recall that the facilitating conditions of this study did not significantly affect user satisfaction with the metaverse platform. It should also be considered that interviewees responded that the metaverse's unique service development is necessary. Fourth, the metaverse must be able to solve problems both in the real world and in the virtual world. The metaverse is not just a created space, it is the space to solve user's problems. The performance expectancy of this study shows the metaverse should be able to solve problems in the real and virtual worlds. For metaverse users to be satisfied, the metaverse must be able to help solve problems in both worlds. In the movie "Ready Player One", which depicts the metaverse well, an unfortunate protagonist in the real world finds vicarious satisfaction in "Oasis", a metaverse platform. People can use virtual-reality technology to treat post-traumatic stress disorder (PTSD) of veterans.

Fifth, the metaverse should be designed to imitate reality. In order for people to easily use the metaverse, the rules applied in reality must be equally applicable to the metaverse. People can feel familiar with this similarity and adapt to the metaverse more easily.

Sixth, the metaverse should be designed intuitively and be easily understood, so that it can be easily used without any difficulties. Effort expectancy in this study described this aspect of user acceptance. To spread the metaverse widely as a media technology, it must be easily available to anyone.

Seventh, the metaverse should support various social interactions. The metaverse should support the symbols we use to interact with others in the real world. A representative example is the avatar's motion-based interaction. We communicate abundantly through various gestures, expressions, and intonations in the real world. These commu-

nications could not only help convey accurate meaning but also increase psychological bonding of the metaverse residents.

Eighth, developers and designers must develop unique functions of the metaverse. Interviewees argued that the current metaverse platform lacks unique functions compared with other information system platforms. In order for the metaverse to have its own market and identity, developers must develop unique functions of the metaverse, such as the immersive mixed reality and the experiential computing supports.

Ninth, regulators should promote the metaverse industry from an ecosystem perspective. The metaverse should not be led by any one company or platform, but by numerous companies, interacting and collaborating with each other. Interactivity between each platform should also be guaranteed. When useful items on one platform become available on another platform, it could facilitate the spread of metaverse platforms.

The tenth–fourteenth topics are discussions on the advantages which companies can benefit from through using the metaverse, and what the roles of the government and regulators are. The dependent variables of this study describe the advantages that companies gain from the spread of the metaverse platform. Ninth, companies can increase their subscribers and expand their influence on the market. Users who are satisfied with the metaverse platform increase their usage intention. Tenth, satisfaction with the metaverse platform increases purchase intention. The more active users are on the metaverse platform, the more willing they are to pay for subscription and item purchases. Eleventh, companies can maximize the network effect due to the increase in subscribers through the spread of positive word of mouth. Twelfth, investment in immersive virtual-reality devices, 3D objects design software, broadband communication networks, and infrastructure is essential to smoothly provide metaverse services. Thirteenth, it is necessary to ease existing regulations to introduce the metaverse. Regulators in Korea defined games as one of the four major causes of addiction and regulated youth's use of games at nighttime. This regulation would not be beneficial for the introduction of the metaverse, which is similar to online games. Rather than regulating for fear of side effects that have not yet occurred, it is necessary to set policy directions that can support various service developments by paying attention to the potential of technology.

In summary, the introduction of the metaverse should be promoted in terms of creating a new world. The metaverse should be able to solve problems both in the real world and in the virtual world. The metaverse should be designed to be easily available to anyone. Unique services of the metaverse should be developed for users. Cooperation of numerous companies and stakeholders within the metaverse industry ecosystem is needed. Companies can make various profits in the metaverse. Regulators should ease existing regulations and create policies to promote the metaverse industry.

## 7. Conclusions

Since I have already specifically explained the meaning of this study in the analysis results and discussion section, this conclusion chapter summarizes the value of this study. Findings of this study could contribute to the following five academic and three practical areas.

There are five academic contributions. First, this study is a pioneering study which investigates a completely new area of the metaverse. The metaverse is considered as Web 3.0, but it is such a new research area that there is no consistent definition. This study provides basic data for subsequential studies by synthesizing the findings of dozens of prior studies to define the metaverse and to organize two flows of research. Second, this study is a theoretical answer to the question of why the metaverse should be introduced. Regarding the claim that the metaverse is a bluff, I argue that the metaverse is an inevitable future change in terms of virtualization of social interaction. Third, this study could successfully explain user acceptance of the metaverse in a unified perspective of the UTAUT as an overlooking theory of this study. Fourth, this study provides clues to new research topics that did not exist before, such as NFTs and cryptocurrency. Fifth, this study expanded the

research scope under the theme of human-oriented values to include a new mixed-reality world by emphasizing the role of social influences, away from the existing research trend that emphasized only the performance of the information system. This study could provide suggestions for how the front-end system should be designed.

There are three practical contributions. First, this study presented several guidelines on how the metaverse should be designed, such as the idea that the metaverse should be able to solve problems both in the virtual and real worlds, should be intuitive and easy to use, and that developers should develop services unique to the metaverse.

Second, I argue that it is urgent to establish a metaverse industry ecosystem. Neither company nor platform can solve all problems which occur in the metaverse. In order for the metaverse to spread widely, an ecosystem in which a larger number of companies play their respective roles is needed. Third, the dependent variable of this study describes the advantages which companies can gain by using the metaverse platform. Companies that have introduced metaverses can effectively attract subscribers by increasing usage intention, purchase intention, and word-of-mouth intention.

Despite these academic and practical contributions, this study has the following limitations. First, "Ifland", the metaverse platform used in this study, has functions which were determined to be too monotonous. This metaverse platform is mainly used through smartphones, and other metaverse platforms support more diverse functions, using immersive virtual-reality headsets. Other metaverse platforms support high-resolution metaverse experiences. Research on user's acceptance and behaviors on various metaverse platforms should be conducted. Second, this study conducted a laboratory experiment following the quarantine rules in Korea, in which more than six people were prohibited from gathering in one place. It is why this study was conducted with a small cohort of 120 people. More persuasive conclusions can be drawn by analyzing more data in subsequent studies. Third, it is necessary to improve the demographic bias, because the respondents for the present study were a group of business students aged 20–30 years. It would be better to conduct research on various demographic groups in the future. Fourth, metaverse technology can increase the media use experience of various media users. The flow experience model, from one of the studies on computer-mediated communication, describes how media affects the various cognitive and affective media use experiences, such as telepresence, focused attention, and flow, in detail. It could be possible to verify how the metaverse affects these dependent variables in future studies. The results will help metaverse user interface designers to design more attractive metaverses.

The results of this study are expected to provide many ideas as basic data for subsequent research on the metaverse, for companies that want to introduce the metaverse as the basis of competitive advantages.

**Author Contributions:** Conceptualization, U.-K.L. and H.K.; methodology, U.-K.L.; validation, U.-K.L.; formal analysis, U.-K.L.; investigation, U.-K.L. and H.K.; resources, U.-K.L.; data curation, U.-K.L.; writing—original draft preparation, U.-K.L.; writing—review and editing, U.-K.L. and H.K.; visualization, U.-K.L.; supervision, H.K.; project administration, H.K. All authors have read and agreed to the published version of the manuscript.

**Funding:** This research received no external funding.

**Institutional Review Board Statement:** Institutional Review Board Statement is not required for this paper in South Korea.

**Informed Consent Statement:** Informed consent was obtained from all subjects involved in the study and written informed consent has been obtained from the subjects to publish this paper.

**Data Availability Statement:** Data sharing not applicable.

**Conflicts of Interest:** The authors declare no conflict of interest.

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
