# Peer review of "UTAUT in Metaverse: An “Ifland” Case"

_jtaer, doi:10.3390/jtaer17020032_

Round 1

Reviewer 1 Report

The research provides a thorough investigation of the metaverse beginning with a detailed analysis of the term followed by a detailed explanation of "Unified Theory of Acceptance and Use of Technology " as it pertains to this study. The methods and methodology are clear and provide insightful data which is analysed well. Overall the paper is presented well but could benefit by improving the use of English grammar.

Author Response

Thank you for your suggestion. This study is an early study to understand metaverse. I did my best to confirm the internal validity of this research. Thank you so much for understanding my efforts by reviewer 1.

Also, this paper had been edited in the fast-track by the professional English editors of American Journal Experts (http://www.aje.com/).

Reviewer 2 Report

The summary, introduction and theoretical framework are very long. They could be shorter as there is redundant information that is shown several times.

In terms of methodology, I think it uses the right tools for analysis and data processing. However, I believe that the main weakness of the study is the number of people in the sample. It is a low number that does not allow us to extract absolute results but rather a first exploration of the object of study.

The "Discussion" section is more a descriptive analysis of the results rather than a comparison with reference studies on which further knowledge is sought. In this section it is necessary to confront the results in order to highlight the findings of the research.

It is an unimportant detail but can be corrected: the name of the singer on line 73 is Travis and not Trevis.

Reviewer 3 Report

The paper presents the idea of empirical verification of the user acceptance of selected metaverse platform by the methodology of Unified Theory of Acceptance and Use of Technology (UTAUT).

The case study focused on the ‘Ifland’ ecosystem is supported by PLS-driven analytics of  students’ opinions (collected in the customized questionnaire). The author also provided in depth analysis of metaverse R&D publications. There are a number of publications on metaverse related subjects, however the author tries to find the commong ground between the user experience design (usability and satisfaction), virtual purchase chain analysis and immersive technolgy as a 'carrier’. The originality of the author’s approach lays in the contribution to enhancing proliferation of metaverse among commercial companies pointing out the ‘natural’ virtualisation of Gen-Z users. Metaverse is characterized by distinctive feature of being totally ‘immersive’ in media, where user is immersed in the virtual/mixed/augmented scenery. On contrary to classic communication channels based on flat image representations, three dimensional stimuli gives the feeling of ‘being there’.

Metaverse may cause potential problems when the system is applied to abstract concepts but works quite well with natural objects/images that are recognized and are familiar to the user. Photorealistic quality is the key factor  to mimic the real-life environment, then.

The suggestions on development of metaverse(s) proposed by the authors is a collection of different scenarions based on immersive technology driven experiences. According to the paper researchers in the field looked primarily for theoretical background (academy) or business models (industry) somewhat omitting the crucial UXD topics. The presented approach shows a valuable method that can help with design process of commercial aspects of the ‘Metaverse’.

Author Response

I sincerely thank you for acknowledging the meaning and value of my paper. Thank you for all your suggestion. I agree to your opinion. I rewrote the fourth comment in conclusion to announcing the needs of future studies to design the more attractive metaverse user interface. (pp. 19, 616-622)
